# Stability of DMPC Liposomes Externally Conjugated with Branched Polyglycerol

**DOI:** 10.3390/ijms23169142

**Published:** 2022-08-15

**Authors:** Alexander A. Beketov, Ivan V. Mikhailov, Anatoly A. Darinskii

**Affiliations:** 1Center for Chemical Engineering, St. Petersburg National Research University of Information Technologies, Mechanics and Optics, 197101 St. Petersburg, Russia; 2Institute of Macromolecular Compounds, Russian Academy of Sciences, 199004 St. Petersburg, Russia

**Keywords:** liposomes, DPD simulations, stability, branched polyglycerol

## Abstract

Vesicles formed by DMPC liposomes externally conjugated with branched polyglycerol-dendrons as well as linear PEG in water solution were simulated using the DPD method. Such a structure of vesicles corresponds to the structure of polymer-grafted liposomes obtained experimentally by the post-insertion method, in which polymer chains are fixed on the outer surface of the liposome. The grafting density, generation number and spacer length of grafted dendrons were varied. It was shown that modification of the outer surface of liposomes due to grafting of hydrophilic dendrons has practically no effect on the size and shape of the vesicle, as well as on the morphology of the lipid membrane up to certain critical thresholds of grafting density, degree of polymerization, and generation number of grafted molecules. Exceeding the threshold values of these structural parameters leads to irreversible deformation of the lipid membrane. Diffusion through the membrane and the transition of grafted molecules from the outer surface of the liposome to the inner surface is not observed for dendrons with a generation number higher than one, even at high grafting densities. The critical values of the generation number and the characteristics of the molecular coating at these values were determined for various grafting densities and spacer lengths of the grafted chains. It was shown that the chemical potential of the grafted dendron can serve as a stability metric for the conjugated liposome. The chemical potential of grafted molecules was calculated using the mean field model of the spherical brush on the liposome surface. An analysis of the simulation data shows that, within the framework of the applicability of the mean field approach, the value of the chemical potential is a sufficient criterion for separating vesicles into stable and unstable forms. These results can be used as a guide for the experimental design of nanocontainers based on lipid vesicles with an external protective coating of branched macromolecules.

## 1. Introduction

Lipid bilayers vesicles known as liposomes are mainly used as transport containers for drugs and genes. The efficiency of drug and gene delivery systems are closely related to their circulation time. However, the circulation time of usual phospholipid vesicles in blood are short due to their detection by the mononuclear phagocyte system (MPS) and subsequent uptake in the liver and spleen [1,2]. It is well known that the recognition, binding and uptake processes occur in a few minutes [3]. The most often used method to increase the circulation time and obtain sterically stabilized (“stealth”) liposomes is modification of the vesicle’s surface by inert polymers. Such a modification prevents the interaction of liposomes with blood components. This decreases MPS cellular uptake as well as an aggregation in blood and increases storage stability of liposomes. One of the most common polymers used for the surface modification is a linear polyethylene glycol (PEG) which is covalently linked to phospholipids [4,5]. However, this polymer (especially the most used methoxy PEG) has a significant disadvantage: the amount of groups for further functionalization is rather small. A good alternative to PEG is the hyperbranched polyglycerol (hbPG) which is highly biocompatible and water-soluble and bears a large share of hydroxyl groups able to be functionalized for directed drug delivery [6,7,8,9]. It was also shown that hbPG demonstrates enhanced protein repulsion compared to PEG [10,11].

There are two methods for the creation of liposomes with their surfaces modified by PEG, hbPG or other hydrophilic oligomers: the pre-insertion and the post-insertion methods [12].

In the first approach, oligomer-containing lipids are added during thin lipid film formation and their hydrophilic blocks become present on both the external and internal surfaces of the lipid bilayer. The hydrophilic shell formed on the internal surface of the liposome does not form a protective coverage, but hinders loading into the liposome, which is a disadvantage. Furthermore, the pre-insertion technique is also ineffective for the preparation of target-specific stealth liposomes with terminal antibodies, antibody fragments and peptides [13,14].

In the post-insertion method, the lipopolymers are slowly added to a diluted suspension of preformed liposomes at temperatures close to the melting point of the constituent lipids [15]. To prevent self-assembly of amphiphilic lipopolymers, their concentration is kept below their critical micellar concentration. Such an approach allows the modification of only the outer liposomal bilayer surface and is more effective in reaching a higher density of the protective oligomeric coating [16]. As a result, the internal space of the vesicle remains available for drugs or genes. Many studies [12,13,14,15] have demonstrated positive outcomes of the post-insertion method over the pre-insertion method. For instance, post-inserted irinotecan-loaded liposomes, showed higher circulation capability than that of their pre-inserted counterparts [15]. It was also shown that intravesicular PEG-lipid degradation was markedly inhibited in the post-insertion method [15].

The structure and functioning of PEGylated and dendronized vesicles have been studied in many experimental works. In addition, the stability of such vesicles was studied with the help of coarse-grained simulations [17,18,19]. In these works, the vesicles obtained by the pre-insertion method were considered with modifying molecules being able to attach from both the outer and inner sides of the lipid vesicle. It has been demonstrated that an increase in the number of conjugated lipids and an increase in the branching of dendron-lipids in the membrane leads to severe deformation and destruction of the vesicles.

In the case of single-side polymer grafting, experimental studies [20,21,22] show that the anchoring of polymers bearing a linear hydrophilic block and grafted hydrophobic anchor groups induces the spontaneous curvature of the liposome membrane. The adsorption of polyampholytes can cause various and sometimes irreversible deformations of liposomes: “necklace” [21], “bud” [22], and “coiling” [20]. To avoid this undesirable effect, it is important to be able to predict the behavior of liposomes depending on the architecture and grafting density of conjugated hydrophilic oligomers.

In this study, we focus on “stealth” liposomes formed by the post-insertion method that provides a purely external hydrophilic coating. The stability of vesicles formed by pure lipids and lipids conjugated with polyglycerol-dendrons is examined. The above-mentioned review shows that this class of lipid-dendron-containing vesicles has wide prospects for use in biomedicine and nanotechnology. The main goal of the present investigation is to determine the critical load of branched polyglycerol oligomers on the outer liposome surface, at which this modified liposome can exist in a dilute aqueous solution. To achieve this goal, the method of dissipative particle dynamics (DPD) [23,24,25,26] was used. This method allows for the evolutionary tracking of liposomes in solutions on sufficiently large temporal and spatial scales. There are several examples of successful applications of DPD for the simulation of lipid membranes and vesicles. In particular, the DPD method was applied to the simulation of cell membrane damage by nonionic surfactants [27], the membrane and vesicle fusion [28] and the mechanism of the bilayer-vesicle transition [29].

## 2. Results and Discussion

### 2.1. Details of Simulations

As a first step, we have prepared a “naked” liposome which serves as a base for the construction of a vesicle with conjugated polymers. As a model, we have chosen the relatively small liposome containing 2314 DMPC. Such liposomes can be prepared experimentally, for example, by sonication of an aqueous suspension of giant vesicles for longer than one hour of sonication time [30]. A sufficiently large cubic periodic simulation box (53 × 53 × 53 nm${}^{3}$) was generated for the creation of the vesicle in the center of the box. The liposome formation was carried out in three stages. First, a pre-assembled pancake-like bilayer encompassing 1167 DPPC lipids on each side and having a thickness equal to 4 nm${}^{3}$ was generated and placed in the simulation box together with 645,686 water beads. After this, the DPD simulation was run in order for the flat lipid bilayer to form a spherical vesicle (Figure 1). In the last stage, the vesicle was equilibrated during the time equal to $10^{6}$ simulation time steps. During this process, the opened pores were closed by fusion of lipids on the surface, releasing extra water molecules from the interior of the liposome, a nematic order of the lipid tails in membrane was formed. There was also a flip-flop of lipids from the concave to the convex surface to compensate for the surface tension on both sides of the vesicle membrane.

The vesicle formation procedure was carried out 10 times starting from different random coordinates of the beads. In a few cases, instead of vesicles, bicelles were formed. These systems were excluded from consideration. In general, it can be argued that a membrane consisting of a given number of lipids forms a spherical vesicle encapsulating an average of 8717 beads of water (26,151 molecules), having 1393 lipid heads on the outer surface and 921 heads on the inner surface. The mean radius of the obtained vesicles is close to 10 nm, which is consistent with the experimental data [30] and the DPD simulation results [31] for the DMPC liposomes.

Modeling of the natural process of post-injection of lipid dendrons into the liposome membrane can take an unreasonably long time. Therefore, an artificial technique was used to construct liposomes with an external hydrophilic molecular coating. The grafted molecular trees were built around the already formed lipid vesicle. Molecules of PEG or polyglycerol-dendrons were randomly grafted onto the lipid heads that comprise the outer surface of the original liposome. The number of grafted chains was varied. The algorithm for constructing molecular structures was organized in such a way that the grafted molecules were distributed as evenly as possible over the entire surface area and all chains were grown on average normal to the surface, forming only the outer coating. In other words, some lipids with heads exposed to the outside of the liposomes were considered as conjugated with PEG or polyglycerol. The number of such conjugated lipids per unit area of the outer surface of the vesicle determines the grafting density. By increasing the number of conjugated lipids, the overall number of beads in the simulation box increased due to the addition of dendrons beads. To keep the total number of beads constant, the corresponding number water beads were eliminated from the box.

To obtain the equilibrium distribution of dendrons over the vesicle surface the following procedure was used. The coordinates of all lipids in the inner layer and water beads inside the vesicle were fixed. That is, in the process of equilibration, the liposome acted as a solid nanoparticle with grafted mobile molecules on the surface, immersed in a mobile solvent. During $5\times 10^{4}$ simulation steps, the molecular coating headed to an equilibrium state, and the gradient of concentration and partial pressure of beads around the immobile vesicle was eliminated. After this, the velocities of all beads were reset to zero and the main simulation was run, during which the vesicle either collapsed or remained “living” for a very long time, assuming its final state. Figure 1 shows the cross-sections of simulation boxes with a pure liposome and a liposome conjugated with branched polyglycerol molecules.

### 2.2. Stability of Conjugated Vesicles

Liposomes with grafted dendrons with a different generation number *g* and spacer length $n_{s}$ were considered. The spacer length was varied from 1 to 10. The grafting density σ was varied from 0.84 nm${}^{-2}$ to 0.08 nm${}^{-2}$ corresponding to the case where dendrons were conjugated with every lipid on the outer surface, with every second lipid, etc., up to every 9th lipid. Every conjugated liposome with the given values of *g*, $n_{s}$ and σ after pre-relaxation were observed for a long simulation time of about 1 million steps, that corresponds approximately to 100μs. If the vesicle was not destroyed during this time, it was classified as stable. If the vesicle membrane ruptured, formed large pores, or even turned into a set of micelles and filamentous aggregates, it was classified as destroyed. Furthermore, a vesicle was considered unstable if at least one lipid-dendron separated from it during this time. Formally, we considered the presence of only two clusters for water particles and one lipid cluster in the simulation box as evidence of vesicle stability. At a fixed grafting density and spacer length, the maximum value of the generation number $g_{max}$ at which the vesicle remains stable was determined. Figure 2 shows a set of cross-sections of dendronized vesicles at $g_{max}$ for different parameters $n_{s}$ and σ.

As can be seen in Figure 2, for stable vesicles any remarkable changes of the shape and size of the original liposome were not observed despite the fact that the layers of dendrons on their surface differ significantly in thickness and density. As a rule, the conjugated dendrons remain on the outer side of the vesicle and do not penetrate into the vesicle. Exclusions are observed only for linear chains and first generation dendrons at high grafting density. For these conjugates, some chains together with their lipids move to the inner side of the vesicle decreasing the effective grafting density on the outer side (see the first column of snapshots in Figure 2). For dendrons of higher generations, such transitions are not observed. This behavior can be explained since massive dendrons cannot pass through the lipid membrane without expending significant energy to form a passage channel.

If for each of the systems shown in Figure 2, the number of generations, spacer length or grafting density are increased, irreversible deformation of the membrane will occur. As an example, in Figure 3, the changes of the morphology of liposomes with grafted linear chains and dendrons of third generation are shown. It can be seen that at the same degree of polymerization *N* the liposomes with a dendron coating are destroyed at lower grafting densities compared to the pegylated liposomes. In addition, as mentioned above, it can be noticed that the grafted linear chains in the case of high grafting density partially penetrate into the inner cavity of the vesicle.

Figure 3 shows diagrams of the values of $g_{max}\left(n_{s},\sigma \right)$ and the values of the molecular load on the liposome surface corresponding to these parameters (the number of monomer units on average per unit surface area).

By using the left diagram in Figure 4, it is possible to predict for the given parameters of grafted chains (σ and $n_{c}$) the maximum generation number $g_{max}$ of conjugated dendrons which will still maintain the vesicle stability. The right diagram in Figure 4 shows that the value of the load on the vesicle surface which corresponds to these vesicles varies widely. This means that this load is not a critical parameter that determines stability of the conjugated vesicles.

### 2.3. Radial Density Distribution

In order to better study the internal structure of conjugated liposomes, the functions of the radial density distribution relative to the center of mass of vesicles were calculated. For comparison, a “naked” liposome, a liposome with grafted dendrons and a liposome with grafted linear chains were considered (Figure 5).

The ordinate axis shows the average numerical density in units of the number of beads per unit volume. The average number density of lipid tails is about 1.5 times higher than the average density of beads in the system. There is also an asymmetry in the distribution of lipid heads. On the inside of the vesicle, the density of the heads is higher than on the outside. Such an asymmetric distribution is compensated by the gain of conformational entropy: in the inner layer of the curved membrane, the tails are less stretched than in the outer layer. A similar behavior of radial density distributions in liposomes was also observed in [32]. The distance from the center to the second peak of the head density (*R* = 11.475 nm) was taken as the outer radius of the vesicle. As the degree of polymerization of grafted chains increases, the density and thickness of the molecular coating increases. With the same degree of polymerization, linear chains form a thicker and looser shell, and dendrons form a less thick and denser shell. This indicates that the density distribution of the membrane components practically does not change its shape even when sufficiently massive chains are grafted to the outer surface. Thus, the surface of the vesicle in a good approximation can be considered as a sphere with a constant radius *R* for all the systems considered, even at a critical value of $g=g_{max}$. That is, there is no gradual deformation of the stretching of the membrane as the load on its surface increases. When a certain critical load is exceeded, irreversible deformation of the membrane occurs abruptly, which in some way resembles the onset of the yield strength for polymer films under load.

### 2.4. Characteristics of the External Coating

One of the main motivations for modifying the liposome surface is to prevent opsonization of the lipid membrane to protect the vesicle from phagocytosis. To do this, the surface of the lipid membrane must be sterically inaccessible to attacking receptors. Therefore, it is important to understand how homogeneous the protective coating is, that is, how much the grafted chains overlap with each other.

The degree of overlap of grafted chains was estimated by the formula:(1)$$X_{over}=\frac{\pi \sigma {\left(RR_{g}\right)}^{2}}{{\left(R+0.5Rg\right)}^{2}}$$
where *R* is the radius of the liposome surface, $R_{g}$—the gyration radius of the grafted dendron in an infinitely dilute solution. This characteristic is zero if the surface of the vesicle is not covered with modifying molecules and is equal to one if the neighboring grafted chains are apart from each other at an average distance of their radius of gyration.

A number of additional simulations were carried out to determine the gyration radii of dendrons. Dendrons without conjugated lipids were placed in separate simulation cells surrounded by water molecules. The stage of relaxation of the system and the stage of recording the equilibrium trajectory, according to which the root-mean-square values of $R_{g}$ were calculated, was about 1000 characteristic relaxation times the square of gyration radius of the dendron in question. The obtained values of gyration radii at critical values of the number of generations in dendrons (corresponding to the diagram in Figure 4), as well as the degree of overlap of grafted dendrons calculated from these values are shown in Figure 6. Based on the presented results, it can be concluded that grafted dendrons overlap poorly at short spacer lengths. The strongest overlap is observed for dendrons with long spacers, but the grafting density should not be too large or too small. Of all the grafted chains considered, at a given liposome size, the most “intertwined” shell is formed by dendrons of the third generation with a spacer length $n_{s}=10$ and a grafting density of 0.21 nm${}^{-2}$.

The obtained values of the gyration radii are well approximated by the power dependence:(2)$$R_{g}∼n_{s}^{\frac{3}{5}}{\left(2^{g+1}-1\right)}^{\frac{1}{5}}{\left(g+1\right)}^{\frac{2}{5}}$$
obtained in the mean-field approximation in [33] for symmetric dendrons in a good solvent (Figure 7). Therefore, to calculate the gyration radii of dendrons with other structural parameters (not considered in this paper), Formula (Equation 2) can be used by applying a numerical coefficient of 0.275.

No less important from the perspective of practical application are such characteristics of the grafted layer as thickness and density (Figure 8). In this paper, the thickness of the hydrophilic molecular coating was calculated as the first moment *h* of the distribution of the end groups relative to the membrane surface. The average density was calculated as the number of monomer units of conjugated molecules related to the volume of a spherical layer enclosed between spheres with radius *R* and R+h.

The thickness of the coating increases with an increase in the length of the spacer of the grafted chains and with a decrease in the grafting density. This is due to the fact that with a decrease in grafting density, it becomes possible to graft more branched dendrons, which form a thicker coating. However, irreversible deformation of dendronized vesicles occurs at lower values of grafting density compared to linear chains with the same degree of polymerization.

The coating density has a more complex dependence on the grafting parameters. It increases with increasing spacer length and grafting density if the number of generations in the dendron is the same (Figure 4 and Figure 8), but the maximum number of generations in the dendron increases with decreasing grafting density. It is worth noting that, as a rule, the densest coating will have a low thickness and vice versa. Highly branched dendrons with a minimum spacer length provide the highest coverage density, but, as can be seen from Figure 6, such dendrons overlap slightly without forming homogeneous brushes.

Another important characteristic is the number of end groups of conjugated molecules per unit area of grafting (see Figure 9), since specific functional groups can be attached to the end groups and subsequent modification of the vesicle for targeted delivery of encapsulated substances. The largest number of end groups per unit surface area of the membrane is observed in the case of short spacers and when the grafted dendrons are most branched. It is worth noting that the highest density of the end groups is observed with the grafting parameters corresponding to the maximum coating density: $n_{s}=1,\;g=6,\;\sigma =0.08$ nm${}^{-2}$ and $n_{s}=1,g=5,\sigma =0.14$ nm${}^{-2}$. It can be concluded that in order to achieve the maximum number of end groups, dendrons with short spacers should be grafted.

### 2.5. Chemical Potential as a Metric of Vesicle Stability

If we consider a liposome with an outer coating of water-soluble molecules as a quasi-equilibrium system, its stability is a consequence of the predominance of hydrophobic interactions of the tails, which tend to minimize the contact area with water and thus maintain a vesicular shape, over entropic forces and repulsion between hydrophilic heads and grafted molecules. On the one hand, if the number of lipids on the outer surface and the shape of the vesicle practically do not change, the same work is spent on removing the lipid from the membrane–the chemical potential of some fixed value ($\mu_{0}$). On the other hand, it can be assumed that the grafted molecules overlap well enough and form a brush on the surface of the vesicle. Thus, in order for the lipid to not leave the membrane together with the conjugated chain, it is necessary for the chemical potential of the chain in the brush to be less than $\mu_{0}$. Thus, the chemical potential can act as a metric of the stability of conjugated vesicles.

The chemical potential of the chain in the brush is the derivative of the free energy with respect to the grafting density:(3)$$\mu =\frac{dF}{d\sigma }.$$

Within the framework of the mean field theory, the free energy of a spherical polymer brush includes two contributions: the free energy of volumetric interactions $F_{int}$ and the free energy of elastic stretching of grafted chains $F_{el}$:(4)$$F=F_{int}+F_{el}.$$

Under the condition of sufficiently strong stretching of the chains, their good overlap and the absence of dead zones for the end groups, the profile of the volume fraction of the polymer brush in a good solvent is described by the parabolic function [34]:(5)$$\phi \left(r\right)=\frac{3\lambda^{2}}{2\alpha^{2}N}\left(H^{2}-r^{2}\right)$$
where *r* is the distance from the grafting surface, *N* is the degree of polymerization of the grafted molecule, *a* is the linear size of the monomer unit, *H* is the thickness of the brush, λ is the topological coefficient, which increases with increasing branching of the chains. The values of the topological coefficient (Table 1) were calculated in [35].

Free energy of volumetric interactions for a spherical brush immersed in a good solvent:(6)$$\frac{F_{int}}{K_{B}T}=\frac{1}{2}\sigma \int_{0}^{H}\phi^{2}\left(r\right){\left(\frac{R+r}{R}\right)}^{2}dr.$$

Free energy of elastic chain stretching
(7)$$\frac{F_{el}}{K_{B}T}=\frac{3\lambda^{2}}{2N^{2}}\sigma \int_{0}^{H}\phi \left(r\right)r^{2}{\left(\frac{R+r}{R}\right)}^{2}dr.$$

The brush thickness is found from the normalization condition
(8)$$\int_{0}^{H}\phi \left(r\right){\left(\frac{R+r}{R}\right)}^{2}dr=N\sigma .$$

Based on Equations (Equation 3)–(Equation 8), the values of chemical potentials for all systems at critical $g_{max}$ (intact vesicles) and their paired values at $g_{max}+1$ were calculated. In Figure 10, the calculated values of chemical potentials are depicted as a function of the degree of polymerization of grafted chains.

Red–pink dots indicate destroyed vesicles and green–yellow dots indicate stable vesicles. Conditionally, these points can be divided by a straight line $\mu =\mu_{0}=16.6\;K_{B}T$. However, there are obvious outliers. They are associated with going beyond the limitation area of the mean-field model. If we remove the points corresponding to the systems in which the flip-flops occurs, as well as the points in which there is a large load on the surface at a sufficiently low density (that is, there is a weak overlap of the chains on the periphery of the brush), the accuracy of separation of stable and destroyed vesicles increases significantly (Figure 10b). Outliers associated with the effect of chain flip-flops are explained by the fact that the effective density of chain grafting decreases due to them and, as a result, the value of the real chemical potential is lower than the calculated value. Thus, the chemical potential calculated using the described algorithm can be considered as a metric for the separation of stable and unstable vesicles taking into account the limitations related to the scope of applicability of the mean-field model.

## 3. Materials and Methods

In the DPD method, the monomer units in polymers and the solvent molecules are replaced by so-called beads, each of which is characterized by a radius, $r_{c}$, and mass, *m*. The pair-wise force describing an interaction between beads *i* and *j* consists of five components: (I) the conservative force ${\vec{F}}_{ij}^{C}$, (II) the dissipative force ${\vec{F}}_{ij}^{D}$, (III) the random force ${\vec{F}}_{ij}^{R}$, (IV) the forces keeping the bonds between linked ${\vec{F}}_{ij}^{bond}$ and (V) the forces maintaining the orientation between consecutive segments in a molecule and thereby ensuring thermodynamic rigidity of spacers ${\vec{F}}_{ij}^{ang}$:(9)$${\vec{F}}_{i}=\sum_{j\neq i}^{N}\left({\vec{F}}_{ij}^{C}+{\vec{F}}_{ij}^{D}+{\vec{F}}_{ij}^{R}\right)+\sum {\vec{F}}_{ij}^{bond}+\sum {\vec{F}}_{ijk}^{ang}$$

Conservative force:(10)$${\vec{F}}_{ij}^{C}=\left\{\begin{array}{cc} a_{ij}\left(1-\frac{r_{ij}}{r_{c}}\right){\vec{e}}_{ij}, & r_{ij}<r_{c}\\ 0, & r_{ij}\geq r_{c} \end{array}\right.$$
where, $a_{ij}$ is the interaction parameter characterizing the maximum repulsion between particles *i* and *j*, $r_{ij}$ is the distance between two particles, ${\vec{e}}_{ij}$ is a unit vector directed along the axis connecting the centers of mass of particles *i* and *j*, ${\vec{e}}_{ij}={\vec{r}}_{ij}/\left|r_{ij}\right|$, and $r_{c}$ is the potential cutoff radius.

Dissipative force:(11)$${\vec{F}}_{ij}^{D}=-\gamma_{ij}\;\omega^{D}\left(r_{ij}\right)\;\left({\vec{\upsilon }}_{ij}\cdot {\vec{e}}_{ij}\right){\vec{e}}_{ij}$$
where $\gamma_{ij}$ is the friction factor, ${\vec{\upsilon }}_{ij}$ is the relative velocity of the particles *i* and *j*, ${\vec{\upsilon }}_{ij}={\vec{\upsilon }}_{i}-{\vec{\upsilon }}_{j}$, $\omega^{D}\left(r_{ij}\right)$ is a switching function describing the change of the friction with distance:(12)$$\omega^{D}\left(r_{ij}\right)=\left\{\begin{array}{cc} {\left(1-\frac{r_{ij}}{r_{c}}\right)}^{2}, & r_{ij}\leq r_{c}\\ 0, & r_{ij}>r_{c} \end{array}\right.$$

Random force:(13)$${\vec{F}}_{ij}^{R}=\sigma_{ij}\;\xi_{ij}\;\omega^{R}\left(r_{ij}\right){\vec{e}}_{ij}$$
where $\sigma_{ij}$ is the intensity of the thermal noise, $\xi_{ij}$ is Gaussian white noise with zero mean and unit variance, $\omega^{R}\left(r_{ij}\right)$ is once again a switching function that describes the decrease of a random force with distance.

Dissipative and random forces play the role of a thermostat. It follows from the fluctuation–dissipation theorem that for thermodynamic equilibrium the following relations must hold [36,37]:(14)$$\omega^{D}\left(r\right)={\left[\omega^{R}\left(r\right)\right]}^{2}$$
and
(15)$$\sigma_{ij}^{2}=2\gamma_{ij}k_{B}T$$
where $k_{B}$ and *T* are the Boltzmann’s constant and absolute temperature, respectively.

Conservative forces are calculated if *i*-th and *j*-th beads are unbonded. In the case of bonded beads, the force ${\vec{F}}_{i}^{bond}$ on particle *i* due to a bond potential $U^{bond}\left(r_{ij}\right)$ is obtained from the general formula:(16)$${\vec{F}}_{i}^{bond}=-\frac{\partial U^{bond}\left(r_{ij}\right)}{\partial r_{ij}}{\vec{e}}_{ij}$$

Forces between three bound particles i,j,k can be described by the angular potential $U^{ang}\left(\theta_{ijk}\right)$ depending the angle $\theta_{ijk}$ formed between those particles
(17)$$\theta_{ijk}=\cos^{-1}\left\{\frac{\vec{r_{ij}}\vec{r_{kj}}}{r_{ij}r_{kj}}\right\}$$

In this case, the force on particle *i* acting in the α direction is given by
(18)$$F_{i,\alpha }^{ang}=-\frac{\partial U^{ang}\left(\theta_{ijk}\right)}{\partial \alpha },\;\;\;\alpha =x,y,z$$

Time integration is performed with a modified version of the velocity-Verlet algorithm [37]:(19)$$\begin{array}{c} {\vec{r}}_{i}\left(t+\Delta t\right)={\vec{r}}_{i}\left(t\right)+{\vec{\upsilon }}_{i}\left(t\right)\Delta t+\frac{1}{2}{\vec{F}}_{i}\left(t\right){\left(\Delta t\right)}^{2},\\ \tilde{\upsilon_{i}}\left(t+\Delta t\right)={\vec{\upsilon }}_{i}\left(t\right)+\lambda {\vec{F}}_{i}\left(t\right)\Delta t,\\ {\vec{F}}_{i}\left(t+\Delta t\right)={\vec{F}}_{i}\left[r_{i}\left(t+\Delta t\right),\tilde{\upsilon_{i}}\left(t+\Delta t\right)\right],\\ {\vec{\upsilon }}_{i}\left(t+\Delta t\right)={\vec{\upsilon }}_{i}\left(t\right)+\frac{1}{2}\left[{\vec{F}}_{i}\left(t\right)+{\vec{F}}_{i}\left(t+\Delta t\right)\right]\Delta t, \end{array}$$
where λ is an empirical factor affecting the stability of the thermostat.

Reduced variables are used in the simulation. The energy is measured in units of $k_{B}T$, the length in units of the cutoff radius $r_{c}$, the time in units of $t_{0}$ and the mass in units of *m*, equal to the mass of one solvent bead. The values $k_{B}T$, *m*, and $r_{c}$ are set to unity. The density of beads $\rho =3r_{c}^{-3}$, the factor λ=0.65 and the time integration step $\Delta t=0.01\;t_{0}$ were chosen.

As a surfactant model, the DMPC lipid molecule conjugated with branched polyglycerol (or linear PEG) was chosen. The examples of chemical structures and corresponding coarse-grained DPD modes are shown in Figure 11B,C. Conjugated lipids were included into the vesicle bilayer consisting of pure DMPC lipids (Figure 11A).

All repulsion potentials are taken to have the same range $r_{c}$ but their amplitudes $a_{ij}$ differ for different bead types. We use four different types of beads: water (W), head (H) and tail (T) beads in lipid molecule and monomer units of conjugated molecule (CM). The interaction parameters between beads were taken from [31,38], they are shown in Table 2. These parameters and the reduced units are mapped in matching with a real bilayer thickness and a measured value for the in-plane diffusion coefficient of lipids. From the experimental data, the thickness of the DMPC membrane is d=3.53 nm [39] and the lipid lateral diffusion coefficient is *D* = 5 μm${}^{2}$/s [40]. Comparing these experimental data and the DPD simulation data for the bilayer lipid membrane ($d=4\;r_{c}$ and $D=0.073\;r_{c}^{2}/t_{0}$), the following physical values were obtained [31]: $t_{0}\approx 11.8$ ns and $r_{c}\approx 0.9$ nm.

The bead friction coefficients $\gamma_{ij}$ were taken also from [31,38] as below:(20)$$\gamma_{ij}=\left\{\begin{array}{cc} 4.5\;k_{B}Tt_{0}/r_{c} & if\;\;a_{ij}<35\;k_{B}T\\ 9.0\;k_{B}Tt_{0}/r_{c} & if\;\;35\;k_{B}T\leq a_{ij}<75\;k_{B}T\\ 20.0\;k_{B}Tt_{0}/ & if\;\;a_{ij}\geq 75\;k_{B}T \end{array}\right.$$

The lipid molecule is represented by the $H_{3}{\left(T_{4}\right)}_{2}$ model according to the lipid model developed by Lipowsky et al. [28,38]. In this model, each *T* bead represents $3.5\;CH_{2}$ groups which implies that one chain contains four beads. The pair-wise interactions concerning lipids–lipids and lipids–water are taken from the abovementioned studies [28,38], as given in Table 2. Two lipid tails, each consisting of four tail beads, are connected with two head beads, while the head group contains three head beads, as shown in Figure 11C. Adjacent beads forming the lipid molecules are connected by the harmonic spring potential:(21)$$U_{L}^{bond}\left(r_{ij}\right)=k_{L}{\left(r_{ij}-r_{L}\right)}^{2}$$
with elastic coefficient $k_{L}=64\;k_{B}T/r_{c}^{2}$, and equilibrium distance $r_{L}=0.5\;r_{c}$. Additionally, the angular potential
(22)$$U_{L}^{ang}\left(\theta_{ijk}\right)=K_{L}\left(1-\cos\theta_{ijk}\right)$$
with $k_{L}=15\;k_{B}T$ is applied to guarantee the stiffness of the lipid tails [28,38].

Such coarse-grained implementation provides a model for the phospholipid DMPC, that reproduces experimentally observed membrane area per DMPC molecule (0.596 nm${}^{2}$ [41]) and the experimentally observed diffusion constant (D=5 μm${}^{2}$/s [40]) for lateral diffusion of DMPC within the flat bilayer membrane.

In our simulations, a part of the lipids was conlugated with branched polyglycerol. Branched polyglycerol is a hydrophilic polymer with a branched dendron-like structure. Each dendron contained $n_{s}$ bonds between branch points (i.e., the length of any of the spacers was equal to $n_{s}$). The total degree of polymerization *N* of one dendron is given by the formula
(23)$$N=n_{s}\cdot \left(2^{g+1}-1\right)$$

A linear chain can be considered as a dendron of zero generation (g=0) with *N* equal $n_{s}$. Figure 12 shows some examples of lipid-dendrons with different number of generations.

The chemical structure of monomer units of branched polyglycerol and linear PEG is almost identical, so to efficiently model conjugated polyglycerol in our DPD simulations, we adopt the coarse-grained model for PEG developed by Lee et al. [42]. In this model, each polyglycerol monomer group has been coarse-grained into one bead (Figure 11). The neighboring beads on each conjugated molecule (CM) are connected by a harmonic bond:(24)$$U_{CM}^{bond}\left(r_{ij}\right)=k_{CM}{\left(r_{ij}-r_{CM}\right)}^{2}$$
where $k_{CM}$ is the bond potential constant, $r_{ij}$ and $r_{CM}$ are the instantaneous and equilibrium bond lengths between branched polyglycerol monomer units, respectively. Here $k_{CM}=2111.3\;k_{B}T$ and $r_{CM}=0.4125\;r_{c}$. The angle potential formed by each triplet of neighboring beads is described by
(25)$$U_{CM}^{ang}\left(\theta_{ijk}\right)=K_{CM}{\left(\cos\theta_{ijk}-\cos\theta_{CM}\right)}^{2}$$
where $K_{CM}=16.4946\;k_{B}T$ is the angle potential constant, $\theta_{ijk}$ and $\theta_{CM}=130^{\circ }$ are the instantaneous and equilibrium angles formed by two consecutive bonds, respectively.

## 4. Conclusions

In the present work, we have simulated the vesicles formed on the base DMPC liposomes conjugated with branched polyglycerol (or linear PEG) in water solution by using the DPD method. The polymer chains were anchored to the outer surface of the liposome, i.e., the structure of the vesicles corresponds to those obtained experimentally by the post-insertion method. It was shown that the grafting dendrons with the number of generation *g* exceeding some critical value $g_{max}$ leads to a loss in vesicle stability. The main result is a prediction of $g_{max}$ of conjugated dendrons at different values of the spacer length $n_{c}$ and grafting density σ. These data can serve as a guide for the experimental construction of lipid vesicles with conjugated dendrons.

From the experimenter’s point of view, it would also be interesting to predict the maximum grafting density $\sigma_{max}$ which can be obtained by grafting dendrons with given *g* and $n_{c}$. Analysis of the diagram in Figure 4 allows for the prediction that $\sigma_{max}$ will decrease both with an increase in *g* and $n_{c}$, but for a more precise estimation of $\sigma_{max}$, additional simulations are necessary. Our results were obtained at a certain value of the liposome radius *R*. What will happen to our predictions if we increase the radius of the vesicle?

For a given dendron structure and grafting density, an increase in *R* leads to an increase in the brush contribution to the free energy of the lipid with conjugated dendron. If we assume that the work on removing the lipid from the membrane $\mu_{0}$ will not change significantly, the condition $\mu =\mu_{0}=16.6\;K_{B}T$ for the chemical potential will be met at lower *g*. This means that grafted liposomes of a larger radius will be destroyed at smaller $g_{max}$. Remembering that the considered vesicles have a size close to the minimal, which can be obtained experimentally, our estimates give an upper bound of $g_{max}$ for real liposomes with grafted chains. To verify these predictions, simulations of vesicles of a larger radius would be desirable.

It would also be interesting to compare the critical grafting densities for vesicles obtained by pre-injection and post-injection methods.

The issues raised form the prospect of continuing this work.

It should also be emphasized once again that when calculating by the DPD method, we used a coarse-grained model (see Figure 11) with interaction parameters for beads (Table 2) obtained earlier for lipids and PEG molecules dissolved in pure water [28,38]. Accordingly, our predictions, strictly speaking, relate to vesicles in a salt-free water solution. However, the main practical goal of the surface modification of vesicles by dendrons is to increase the stability of liposomes dissolved in the blood, which contains salt ions. The question then arises: To what extent do our results relate to PEG-conjugated vesicles in the blood? For a strict answer to this question, it is necessary to parameterize the interactions between beads in our model based on results of MD simulation and experimental data for DMPC liposomes in physiological conditions. To date, such parameterization has not been carried out. However, we can try to make some predictions about the effects of salt on the stability of dendronized vesicles. Hydrophilic interactions between the heads of lipid vesicles are largely due to the presence of charged groups in them. The addition of salt reduces the effect of electrostatic interactions and, accordingly, decreases the repulsion between beads of the heads. As a result, one has to expect an increase in chemical potential $\mu_{0}$ which characterizes the work necessary to “pull out” the lipid molecule from the vesicle. Therefore, a vesicle of the same size dissolved in blood will keep its stability with a larger dendron load on its surface than in pure water. This means that our results predict the lower bound for values of maximum generation number $g_{max}$ of conjugated dendrons for liposomes with a given size and PEG-conjugated vesicles stable in pure water will remain stable in the blood.

## Figures and Tables

**Figure 1 ijms-23-09142-f001:**
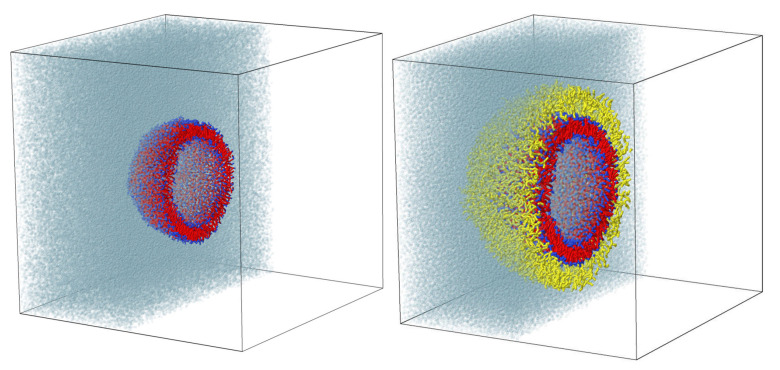
Cross-sections of the simulation boxes with a pure liposome (**left**) and a liposome conjugated with branched polyglycerol (**right**).

**Figure 2 ijms-23-09142-f002:**
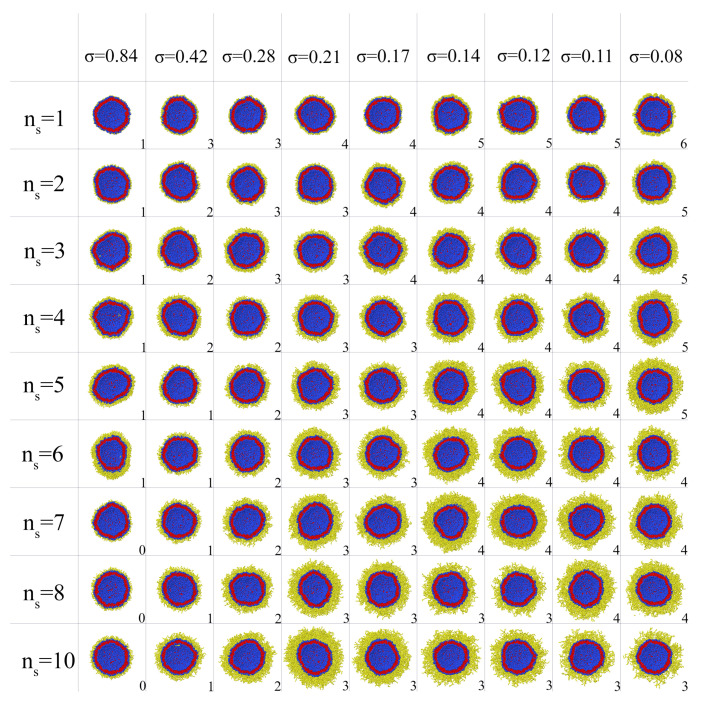
Snapshots of vesicles with grafted dendrons at different lengths of spacers $n_{s}$ and grafting density σ. The systems at critical generation number $g_{max}$ of dendrons are shown. The values of $g_{max}$ are indicated by numbers in this figure.

**Figure 3 ijms-23-09142-f003:**
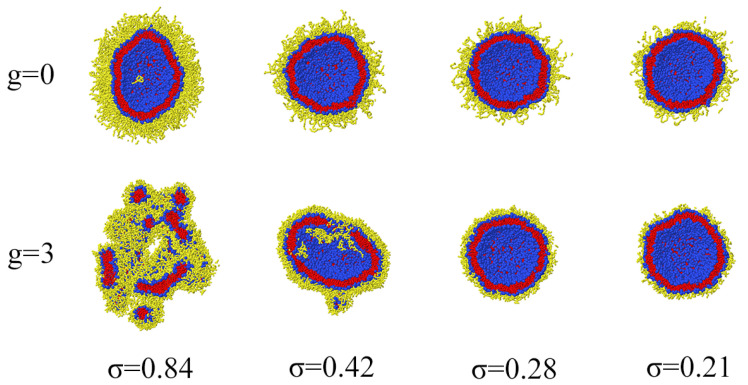
Cross-sections of liposomes with grafted linear chains and third-generation dendrons at different grafting densities. For all cases N=28.

**Figure 4 ijms-23-09142-f004:**
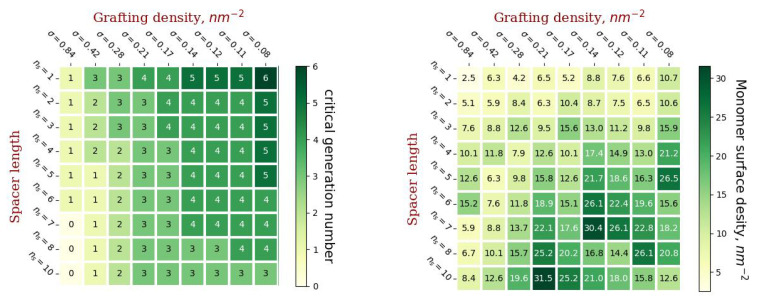
Diagrams illustrating critical values for the number of generations of grafted dendrons at various parameters $n_{s}$ and σ and corresponding to these parameters loads on the surface.

**Figure 5 ijms-23-09142-f005:**
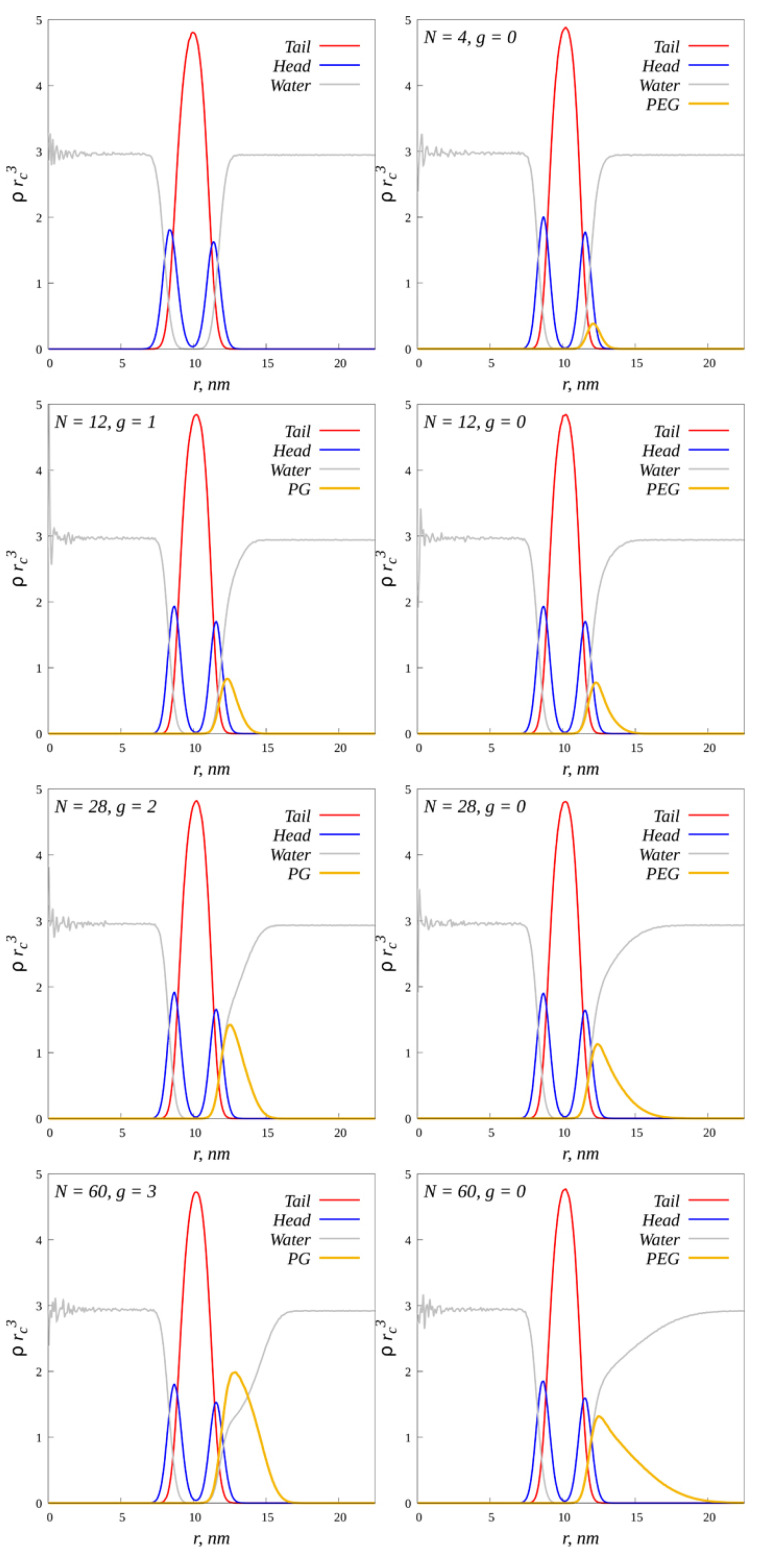
Radial density distribution of various DPD beads relative to the center of mass of vesicles, red indicates lipid tails, blue–heads, gray–water, yellow–grafted molecules.

**Figure 6 ijms-23-09142-f006:**
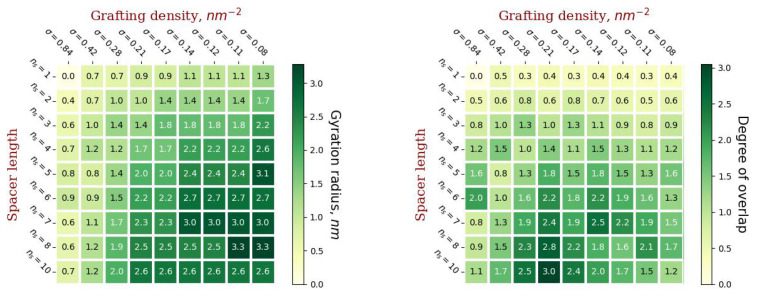
The gyration radii of single dendrons at critical values of the number of generations (**left**) and the degree of overlap of grafted chains (**right**).

**Figure 7 ijms-23-09142-f007:**
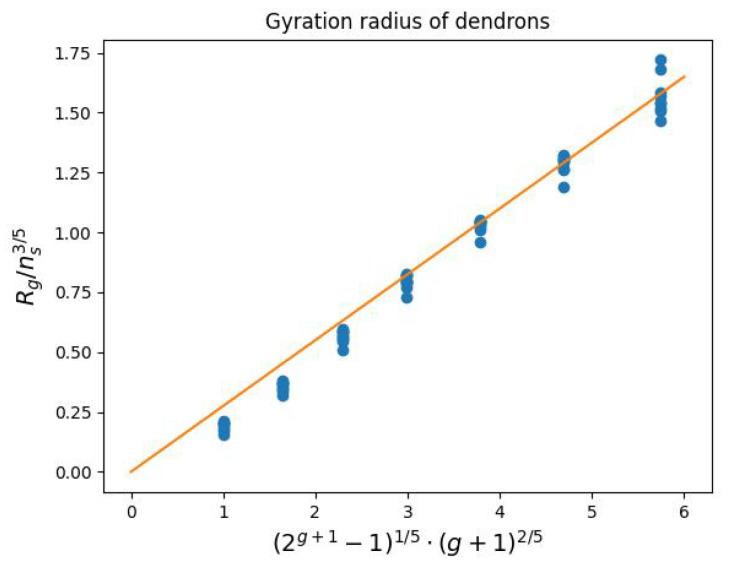
Approximation of the dependence (Equation 2) based on the dendron gyration radii calculated from the simulation (slope 0.275).

**Figure 8 ijms-23-09142-f008:**
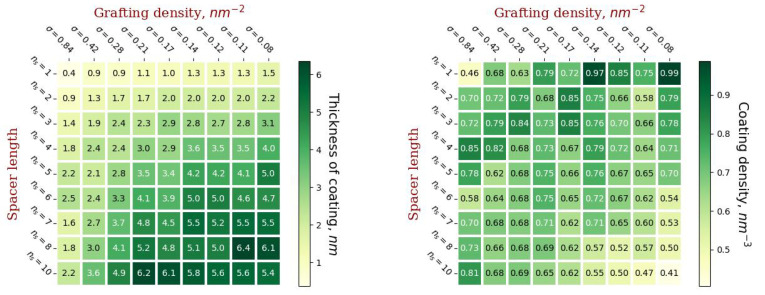
The thickness of the formed molecular coating (**left**) and its density (**right**).

**Figure 9 ijms-23-09142-f009:**
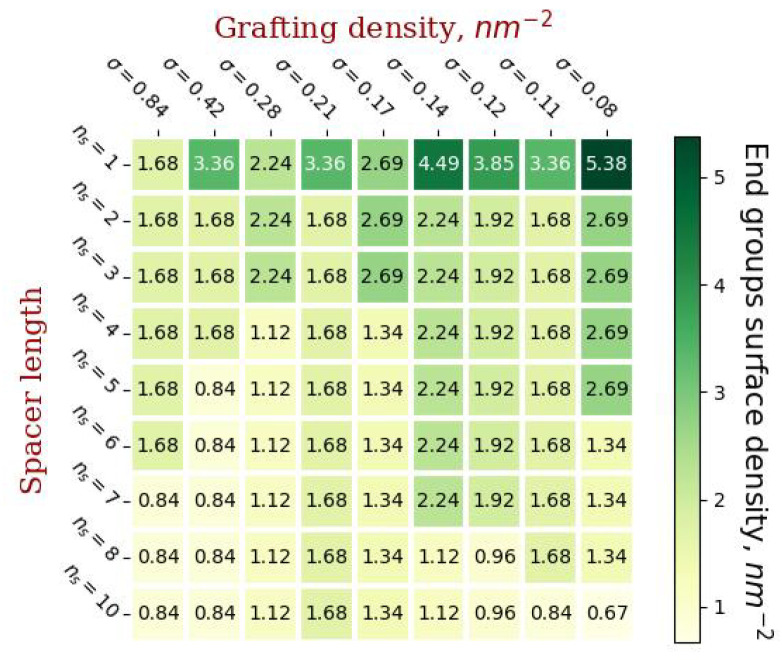
Density of the end groups per unit surface area.

**Figure 10 ijms-23-09142-f010:**
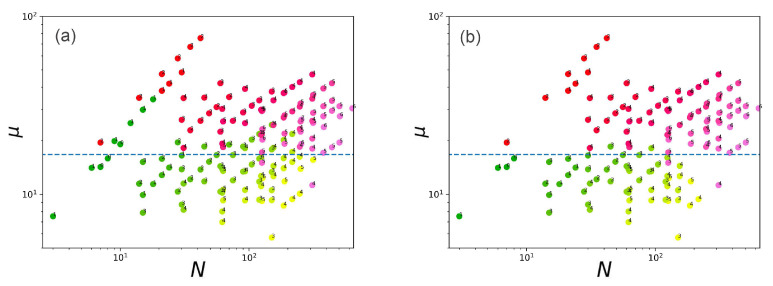
Chemical potentials of chains in a spherical brush calculated in the mean-field approximation as a function of the degree of polymerization of grafted chains in units $K_{B}T$. The gradient of color from green to yellow shows stable vesicles, from red to pink—destroyed ones, numbers indicate the number of generations of dendrons. The color gradient shows an increase in grafting density. In graph (**b**), in contrast to (**a**), the outliers of some false-negative points have been removed.

**Figure 11 ijms-23-09142-f011:**
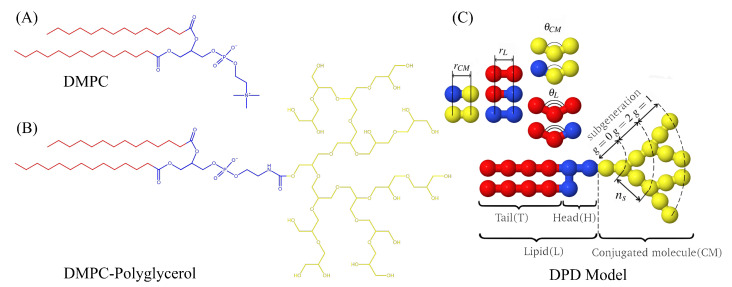
The pure DMPC lipid (**A**), the DMPC lipid conjugated with branched polyglycerol (**B**) and DPD model of this lipid (**C**) conjugated with a second generation polyglycerol-dendron (g=2) with a spacer length equal to $n_{s}=2$ for this example.

**Figure 12 ijms-23-09142-f012:**
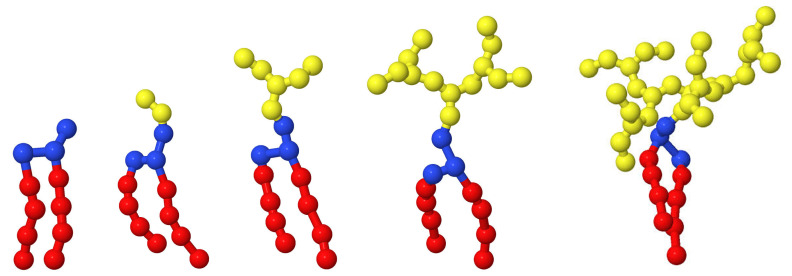
The coarse-grained models of the pure lipid (left) and lipids conjugated with dendrons with a spacer length equal to 2 and generation numbers from 0 to 3.

**Table 1 ijms-23-09142-t001:** Topological coefficient.

*g*	0	1	2	3	4	5	6
λ	1.571	1.231	1.020	0.844	0.693	0.563	0.453

**Table 2 ijms-23-09142-t002:** Interaction parameters, $a_{ij}$, between beads *i* and *j*, in units of $k_{B}T/r_{c}$. Letters *W*, *H*, *T* and CM represent water molecules, hydrophilic lipid heads, hydrophobic lipid tails and monomer units of conjugated molecules (branched polyglycerol-dendrons or linear PEG), respectively.

	*W*	*H*	*T*	*CM*
*W*	25.0	30.0	75.0	26.3
*H*	30.0	30.0	35.0	26.3
*T*	75.0	35.0	10.0	33.7
CM	26.3	26.3	33.7	25.0

## Data Availability

Not applicable.

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
