# Peer review of "Stability of DMPC Liposomes Externally Conjugated with Branched Polyglycerol"

_ijms, 2022, doi:10.3390/ijms23169142_

Round 1

Reviewer 1 Report

The manuscript “Stability of DMPC Liposomes Externally Conjugated with Branched Polyglycerol” by Beketov et al. is a very interesting simulative study on vesicle formation.

The whole manuscript is well written. The references are balanced and the state of the art and the physical fundamentals are well explained. The findings are discussed with literature and the methods are very well documented.

I like this contribution which is a impactful contribution to the biophysical field of liposome formation.

There are only a few technicalities which need to be improved.

The abbreviations for DMPC, DPD and PEG are not explained in the abstract (some are not explained at all)

The text such as legends and axis labelling cannot be read inf figures 6-12 with figure 7 being the most challenging one. Please increase the font size in these figures.

Author Response

Thank you very much for your review and comments. 1. Abbreviations, according to the rules of the journal, are indicated at the end of the manuscript. 2. For better readability, figures 6-11 were enlarged in size, in figure 12 the labels along the axes were enlarged.  

Reviewer 2 Report

The authors have reported the calculational predictions of liposomes composed of lipids tethered to various sizes of PEG dendrons in a water environment. These results will be helpful and informative for researchers in the field of materials science and materials chemistry for drug delivery of liposomes. Whereas the reviewer thinks that the authors’ study in this manuscript is quite interesting, suggestive, and well-organized, some descriptions are not enough. The authors’ manuscript is not suitable for publication in “International Journal of Molecular Science” in the present form.

From these considerations, the reviewer recommends to accepting for publication in " International Journal of Molecular Science," if the following issues are resolved.

(1)   This coarse, bold, yet still appropriate computational approach predicts a threshold for liposome shape stability for dendron structures, which is very interesting and suggestive. The authors made their computational predictions in water, but the environment in the blood is not only water but also some salts. The presence of salt is thought to affect the intermolecular interactions of lipids and the stability of liposomes. What is the effect of salt concentration on liposome shape stability in this calculation method? Maybe this calculation will be the next job, but is it possible to have an estimate of its impact?

Author Response

As a response to the remark of the reviewer 2  we have added the following text to the end of the conclusion section. (lines 365-384)

It should also be emphasized once again that when calculating by the DPD  method, we used a coarse-grained model (see Fig.1) with interaction parameters for beads (Table 1) obtained earlier for lipids and  PEG  molecules dissolved in pure water..Accordingly, our predictions, strictly speaking, relate to  vesicles  in the salt free  water solution. However, the main practical goal of the surface modification of vesicles by dendrons is to  increase the stability of  liposomes  dissolved in the blood, which contains salt ions. The question arises to what extent our results relate to PEG conjugated vesicles in the blood? For a strict answer to this question, it is necessary to parameterize the interactions between beads in our model based on results of MD simulation and experimental data for DMPC liposomes in the physiological conditions. So far, such parameterization has not been carried out. However, we can try to make some predictions about the effects of salt on the stability of dendronized vesicles. Hydrophilic interactions between the heads of lipid vesicles are largely due to the presence of charged groups in them. The addition of salt reduces the effect of electrostatic interactions and, accordingly, decreases  the repulsion between beads of the heads. As a result one has to expect an increase of chemical potential Ď»0 which characterizes  the work which is necessary to " pull out" the lipid molecule from the vesicle. Therefore the vesicle with the same size dissolved in blood will keep its stability at larger dendronic load on its surface than in pure water.. It means that our results  predict  the lower bound for values of maximum generation number gmax of conjugated dendrons for liposomes with a given size and PEG conjugated vesicles stable in pure water will remain stable in the blood.